# Meta-review of the barriers and facilitators to women accessing perinatal mental healthcare

Rebecca Webb [1], Nazihah Uddin,[1] Georgina Constantinou,[1] Elizabeth Ford [2], Abigail Easter [3,4] Judy Shakespeare [5] Agnes Hann,[6] Nia Roberts,[7] Fiona Alderdice,[8] Andrea Sinesi,[9] Rose Coates,[1] Sally Hogg,[10] Susan Ayers,[1] The MATRIx Study Team

For numbered affiliations see end of article.

**Correspondence to**
Dr Rebecca Webb;
rebecca.webb.2@city.ac.uk

## ABSTRACT

Perinatal mental health (PMH) problems are common and can have an adverse impact on women and their families. However, research suggests that a substantial proportion of women with PMH problems do not access care.

**Objectives** To synthesise the results from previous systematic reviews of barriers and facilitators to women to seeking help, accessing help, and engaging in PMH care, and to suggest recommendations for clinical practice and policy.

**Design** A meta-review of systematic reviews.

**Review methods** Seven databases were searched and reviewed using a Preferred Reporting Items for Systematic Reviews and Meta Analyses search strategy. Studies that focused on the views of women seeking help and accessing PMH care were included. Data were analysed using thematic synthesis. Assessing the Methodological Quality of Systematic Reviews-2 was used to assess review methodology. To improve validity of results, a qualitative sensitivity analysis was conducted to assess whether themes remained consistent across all reviews, regardless of their quality rating.

**Results** A total of 32 reviews were included. A wide range of barriers and facilitators to women accessing PMH care were identified. These mapped across a multilevel model of influential factors (individual, healthcare professional, interpersonal, organisational, political and societal) and across the care pathway (from decision to consult to receiving care). Evidence-based recommendations to support the design and delivery of PMH care were produced based on identified barriers and facilitators.

**Conclusion** The identified barriers and facilitators point to a complex interplay of many factors, highlighting the need for an international effort to increase awareness of PMH problems, reduce mental health stigma, and provide woman-centred, flexible care, delivered by well trained and culturally sensitive primary care, maternity, and psychiatric health professionals.

**PROSPERO registration number** CRD42019142854.

## INTRODUCTION

Perinatal mental health (PMH) problems commonly consist of anxiety disorders, depression, post-traumatic stress disorder and stress-related conditions such as adjustment disorder. They can also include more severe difficulties such as postpartum psychosis, and many PMH problems are comorbid.[1 2]

PMH problems can adversely impact women and their families. They are associated with obstetric physical health complications, such as increased risk of pre-eclampsia, antepartum and postpartum haemorrhage, placental abruption, stillbirth,[3–5] and preterm birth.[6 7] Furthermore, suicide is a leading cause of death during the perinatal period in higher-income countries, accounting for 5%–20% of maternal deaths.[2 8 9] Perinatal suicide accounts for between 0.655% and 3.55% of pregnancy-related deaths in lower-middle-income countries (LMICs).[10] Research has also found PMH problems are associated with a child's cognitive, language[11–14] and behavioural development.[13 15 16] PMH problems may also mean a woman's child is at an increased risk of developing mental health difficulties themselves.[17–19] Furthermore, PMH problems can impact on a woman's relationships with her partner, such as by a decline in relationship satisfaction,[20] increased strain on the couple relationship[21 22] and relationship breakdown.[23] There is also a large cost to society and healthcare services, with PMH problems costing the UK approximately £8.1 billion every year.[24]

Evidence-based PMH care can reduce the negative impacts on women and their

families. For example, cognitive–behavioural therapy,[25] psychological therapies[26] and certain antidepressant medications[27] have been shown to be effective in reducing PMH symptoms.

Globally, evidence-based guidelines exist for PMH care. The WHO Millennium Development Goal 5 is to improve maternal health,[28] and states that a mental health component should be incorporated as an integral part of maternal health policies, plans and activities in all countries.[29] However, research suggests access to PMH care is variable[30–33] with only 30%–50% of women with PMH problems identified, and less than 10% referred to specialist care.[34–36] This variable access could be due to multiple reasons, such as difficulties with implementing PMH services,[37] or due to barriers experienced by women.

Multiple systematic reviews have explored barriers and facilitators to women accessing PMH care. Each systematic review varies slightly in relation to its aim and methods making it hard to extract the information needed to design PMH services in a more accessible way. A systematic review of systematic reviews, or a meta-review, is arguably the most suitable way to synthesise results by combining results from multiple reviews into a single body of evidence. This allows for comparison of results from multiple reviews. A meta-review would make it easier for healthcare providers and policymakers to access the information and use it to inform their decisions.[38 39] Therefore, the primary aim of this research is to determine the key barriers and facilitators to women deciding to seek help, access help and engage in PMH care using a meta-review.

## METHOD

The protocol for this review has been registered on PROSPERO (CRD42020193107) (see online supplemental appendix 1).

### Patient and public involvement

This project was developed with patient and public involvement representatives from the NCT in England, and the Maternal Mental Health Change Agents, a group of women with lived experience of PMH problems in Scotland.

### Data sources and searches

Searches were carried out by NR in CINAHL (1982–present); Embase (1974–present); Medline (1946–present); PsycINFO (1806–present), Cochrane, SCOPUS and Turning Research into Practice Medical Database. Searches were completed on 4 August 2021 and forward and backward searches were completed by 8 September 2021. See online supplemental appendices 2 and 3 for full search syntax and results.

### Study selection

Reviews were included if they used a Preferred Reporting Items for Systematic Reviews and Meta Analyses[40] search strategy and focused on the views of women seeking help and accessing care for perinatal mental illness. See online

supplemental appendix 4 for full inclusion criteria. Search results were imported into EndNote and duplicates and papers not meeting initial inclusion criteria (foetal distress, oxidative stress and non-English papers due to translation times and costs) were removed by NR. The remaining studies were imported into Eppi-Reviewer V.4, where results were double screened by title and abstract by two people (RW and GC). Following this, full-text screening was carried out by two people (RW and GC).

### Data collection process and data items

Data extraction was carried out using Microsoft Excel by RW. Double coding of extracted data was carried out for a proportion of included reviews (n=3, 10%) by GC.

### Critical appraisal of reviews

Methodology sections of included systematic reviews were appraised using the Assessing the Methodological Quality of Systematic Reviews-2.[41] A decision was made to include reviews where confidence in results was evaluated as low and critically low because these reviews focused more on marginalised women, such as refugees, migrants, women with a low income and women living in LMICs, to ensure the experiences of these seldom-heard women were captured. To improve the validity of results, a qualitative sensitivity analysis was carried out to assess whether themes remained consistent across all reviews, regardless of their quality rating (see online supplemental appendices 5–8).

### Synthesis of results

Results were analysed by RW using a thematic synthesis[42] in NVivo and Microsoft Excel. Themes were mapped onto a multilevel framework adapted from Ferlie and Shortell's Levels of Change framework (individual level, group/team level, organisational level and larger system/environment level)[43] and used in a previous systematic review on barriers and facilitators to implementing PMH care, carried out by the review authors.[37] The levels identified in the previous review reflect the reviewed literature and the complexities of the health services and are as follows: individual, health professional (HP), interpersonal, organisational, political and societal. These will be described in more detail below. The mapping of descriptive themes was developed deductively from the initial theoretical framework and then inductively revised as new themes emerged. The mapping of descriptive themes was discussed by all review authors before being finalised. Differences of opinion were resolved through discussion. Recommendations were developed for policy and practice based on the most cited themes. For a more detailed methodology, please see online supplemental appendix 5.

## RESULTS
### Review selection and review characteristics

Screening identified 32 reviews to be included in the meta-review (see figure 1). See online supplemental appendices 9 and 10 for review characteristics.

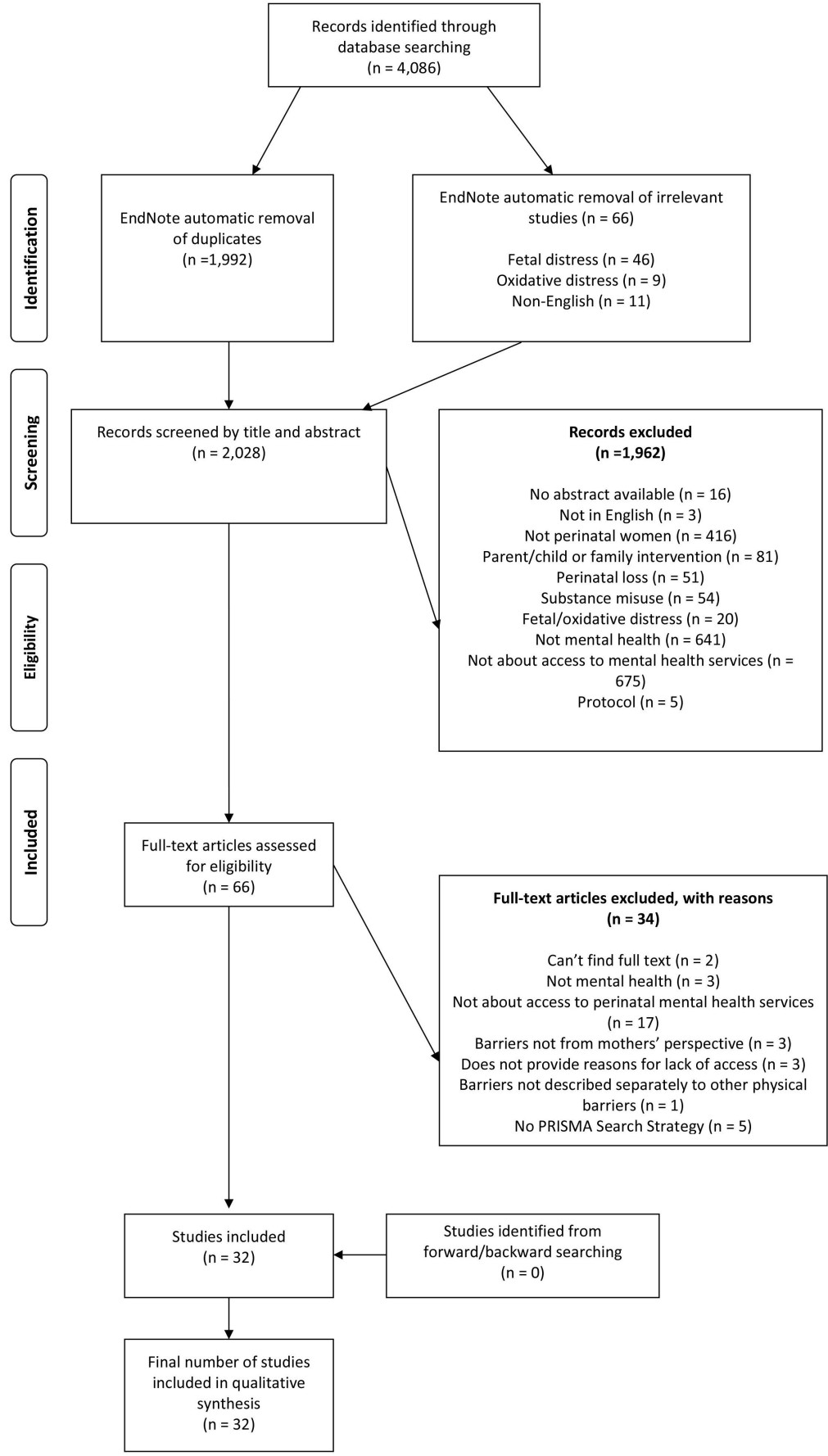

**Figure 1** PRISMA flow diagram. From Moher et al.[94] PRISMA, Preferred Reporting Items for Systematic Reviews and Meta Analyses.

### Risk of bias within studies

Most reviews were evaluated as having low (n=14) or critically low (n=5) confidence with their results. The remainder had moderate (n=8) or high (n=5) confidence (see online supplemental appendix 11).

### Synthesis of results
#### Determining the barriers and facilitators to women help-seeking and accessing PMH care

A total of six overarching themes, mapped onto a multi-level framework,[43] made up of 62 subthemes were identified (see online supplemental appendix 12). The multilevel framework is an extension of Ferlie and Shortell's Levels of Change framework[43] with six levels, instead of four. The first level is the individual level, which reflects factors related to the person themselves. The second level is HP, which reflects factors related to the HP. Interpersonal refers to the relationship between women and HPs, this is an extension of Ferlie and Shortell's work and was included because this theme was represented in the literature.[37] The next theme is organisational, which relates to how the organisation is run, and the type of care the organisation delivers. The literature provided multiple examples of how women wanted their care designed. As the organisation is in charge of designing and providing care, ideal care was mapped as a subtheme under this theme. The political level relates to the policies and governing that may impact on women and healthcare. The societal level relates to larger societal factors, such as stigma. It is important to note that these levels do not exist in isolation but often impact one another, for example, a lack of political funding and policy will have a negative impact on how an organisation is run, staff burn-out and thus the care delivered to women.

Each level of the mult-level framework (figure 2) maps on to at least one part of the care pathway (figure 3). Each level of the multilevel framework will be outlined below, and within each level, the most cited barriers and facilitators will be presented following the chronology of the care pathway outlined in figure 3. Recommendations for practice and policy can be found in table 1. It should be noted that the review draws on international evidence, and not all the factors identified will exist to the same extent in all places.

### Individual-level factors

Individual-level factors were identified by 25 reviews.

#### Deciding to consult

Barriers that prevented women from deciding to consult included, not understanding the role of HPs (n=6), and not knowing what perinatal mental illness is (n=14):

> I don't know what postnatal depression is—how you're supposed to feel, look, or whatever. I don't know. I have no idea … what exactly is postnatal depression? What are you supposed to be doing, saying, or whatever? I don't know. (44, p.e694)

Not knowing what perinatal mental illness is led to some women believing their symptoms were a normal part of motherhood (n=8), or led to women attributing

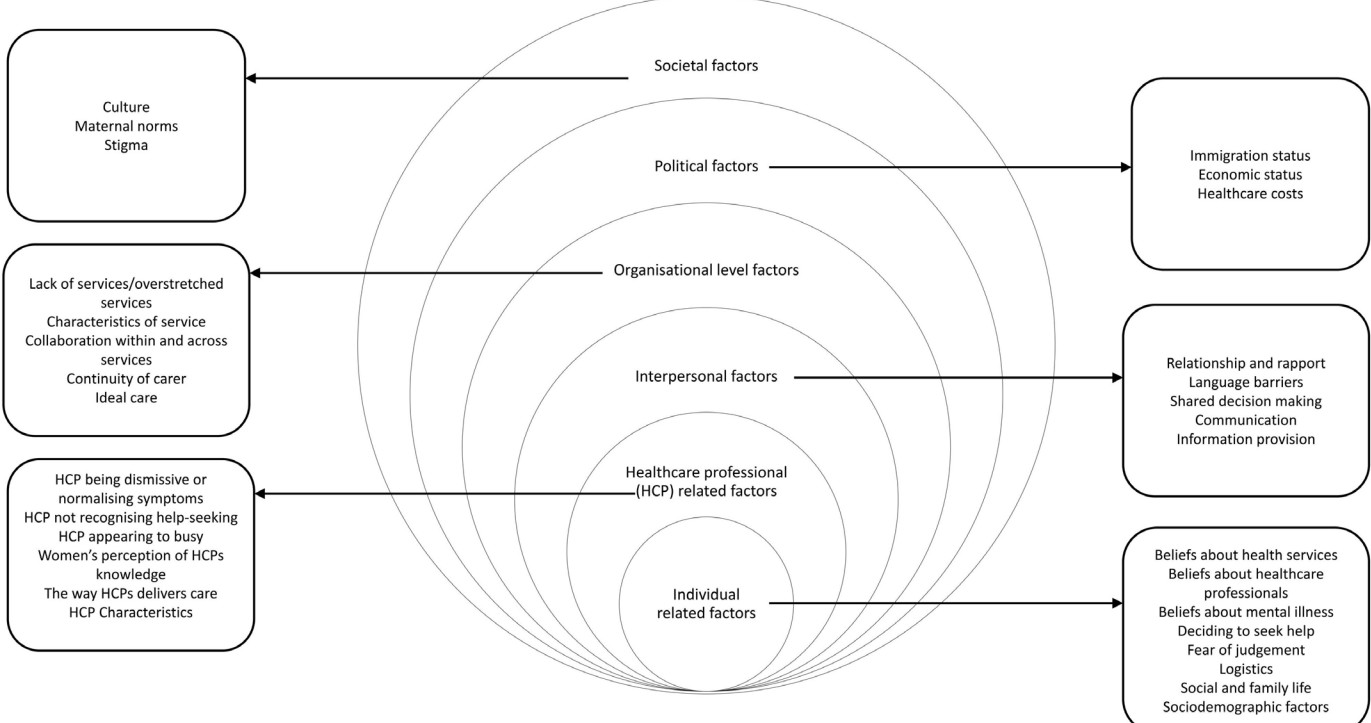

**Figure 2** The MATRIx multilevel model of barriers and facilitators to women accessing PMH care. PMH, perinatal mental health.

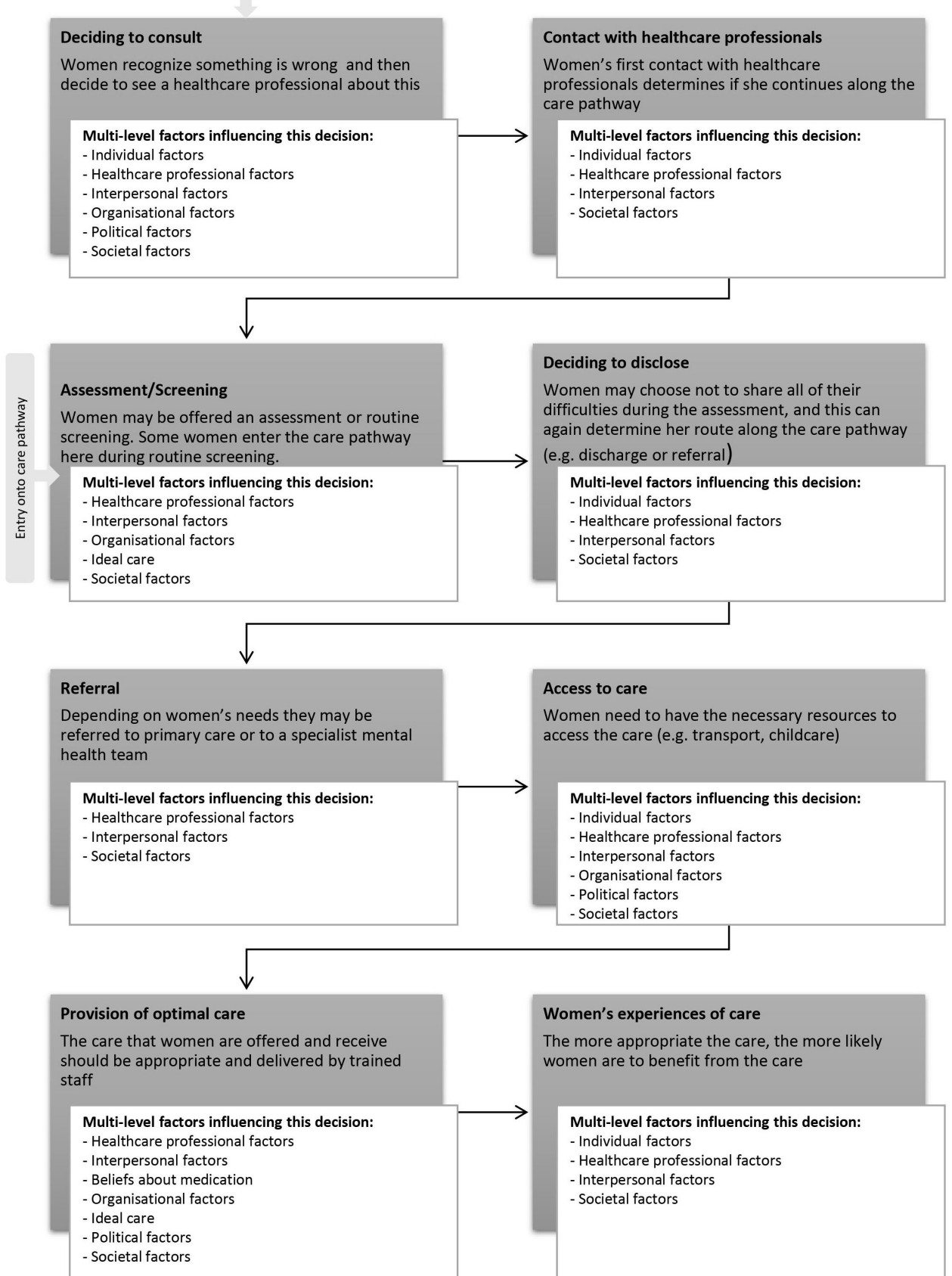

**Figure 3** Barriers and facilitators mapped onto the MATRIx care pathway. Note: Some parts of the pathway are redundant in healthcare systems where the woman can contact mental health services directly (eg, France or via Improving Access to Psychological Therapies services in the UK). Further, the process is not always linear women might jump over certain stages.

**Table 1** Recommendations for improving PMH care for women

| System-level factor | Theme | Recommendation |
|---|---|---|
| Individual | Beliefs about health services<br>Beliefs about HPs<br>Beliefs about mental illness<br>Fear of judgement<br>Logistics | Improvement of mental health literacy for, women, family, friends and all who meet perinatal women*†<br>Free access to healthcare‡<br>Woman-centred care§ |
| Healthcare professional | Characteristics<br>Time<br>Training and knowledge | Attend training in communication skills**<br>Attend training in PMH to reduce stigma**<br>Attend training in cross-cultural presentations of mental health difficulties** |
| Interpersonal | Relationship and rapport<br>Language barriers<br>Shared decision making<br>Communication<br>Information provision | Healthcare professional to attend training in communication skills**<br>Healthcare professional to attend training in PMH to reduce stigma**<br>Healthcare professional to attend training in cross-cultural presentations of mental health difficulties¶**<br>Provision of continuity of carer¶ |
| Organisational (including ideal care) | Lack of services/overstretched services<br>Characteristics of the service including continuity of carer<br>Collaboration across services | Individualised and culturally appropriate care co-designed with women.¶<br>Improved funding for PMH services.‡<br>Improved guidance for implementing PMH care*†† |
| Political | Immigration and economic status<br>Healthcare costs | Equal rights to healthcare*<br>Free healthcare*<br>Laws to protect those with immigration status*<br>International policy that supports the funding and implementation of personalised culturally appropriate care¶ |
| Societal | Stigma<br>Culture<br>Maternal norms | International, culturally sensitive public mental health campaigns to increase knowledge about mental illness and improve attitudes about people with mental illness[76 95–99]¶<br>The continuation of international policies to promote gender equality, higher paid parental leave,[65] increased opportunity for women in the labour force,[66 67 100] the right to access contraception and abortion.[68]* |

*Recommendations for public health services (eg, the NHS, the European Public Health Association, Public Health Association of Australia).
†Recommendations for third sector organisations (eg, the National Childbirth Trust, UK; The Babes Project, Australia).
‡Recommendation for the government.
§Recommendation for organisation.
¶Recommendations for healthcare professionals.
**Recommendations for implementing PMH assessment, care and treatment can be found in Webb et al.[37]
††Recommendations for academics/researchers.
HP, health professional; NHS, National Health Service; PMH, perinatal mental health.

their symptoms to external causes (eg, job loss; n=8), or physical causes such as hormones (n=9):

> I thought it was just lack of sleep and this heavy cold. I thought that after a good night's sleep it would get better, and I would be able to manage (44, p.e696)

Other barriers at this stage of the care pathway included dealing with symptoms by ignoring them (n=6), or minimising them (n=12); not knowing where to go in order to seek help (n=7); and the fear of being seen as a bad mum or fear of social services involvement (n=7).

Facilitators to deciding to consult was recognising that something was wrong (n=9) and having supportive family and friends (n=5):

> That's when I thought, you know: "Something is really wrong here, I need to go to the doctors if

I'm thinking about killing myself." (44, p.e694) [Recognising something is wrong]

> It was sort of my partner saying to me: "Right, if you don't go, I'm basically making you an appointment … You can't just keep feeling like this." (44, p.e694) [Supportive family and friends]

### Deciding to disclose

One barrier at this stage of the care pathway was not understanding the HP's role, perceiving them as agents of social control (n=4):

> 'I don't really know what their job is. Nobody gave me, like, the parameters of this role of the health visitor [maternal and child health nurse]….' (44, p.e695)

Linked to this was the fear of social services involvement and the removal of their child (n=7), as well as fears of being judged to be a bad mum (n=8):

> I even went in at 3 months and I talked to a health nurse, and I just lied through my teeth because I thought, what are they going to do if they find out I can't be a good mom? (45, p.732–733)

### Access to care

The most cited barrier at this stage of the care pathway was logistical reasons (n=13) such as travel costs, lack of childcare and timing of services.

### HP-related factors

HP-level factors were reported by 18 reviews.

### First contact with HPs

HPs not recognising women's help-seeking or symptoms (n=4), and being dismissive or normalising women's symptoms (n=8) were barriers at this early stage in the care pathway:

> I did ask for support, but I didn't really get any. And the health visitor's response—"Well you seem like you're doing all right"—which kind of closes it off, doesn't it (44, p.e696)

Linked to this, HPs appearing to not have enough time to address women's concerns was also a barrier

> The health visitor said something like: "You know, in this community we have to look after a thousand and something babies." And that instilled in me the feeling, like: "Oh, they are very busy these people, and I don't have to be bothering them all the time (44, p.e696)

### Assessment/screening

Assessment being carried out in a formulaic tick-box way, or not being carried out at all (n=3) was the most cited barrier.

### Deciding to disclose

The most reported barrier at this stage of the care pathway was HPs appearing to not have enough time (n=4) or HPs being dismissive or normalising women's symptoms (n=4).

### Referral

Women's perception of HPs knowledge of referral pathways/other services (n=3) and HPs not recognising women's help-seeking or symptoms (n=2) were barriers to referral:

> I purposely circled the things 'cos I'm struggling … the health visitor didn't get back to me, which I'm really disappointed about. (44, p.e696)

### Access to care, provision of optimal care and women's experiences of care

These stages of the care pathway were mainly influenced by the characteristics of HPs. For example, HPs who were trustworthy, responsive, non-judgemental, understanding, caring, interested, warm, empathetic and positive (n=12) were facilitators. On the other hand, unhelpful or uninterested staff were barriers (n=2).

### Interpersonal factors

Interpersonal-level factors were identified by 14 reviews.

### Deciding to consult, deciding to disclose and women's experience of care

The development of a strong and trusting relationship with a HP (n=10) was a facilitator to women at each of these stages of the care pathway:

> She's a supplement to my own mother. She's easy to talk to. I depend on her. She's not just there to take care of the baby but for the mothers too. She started a group for us new mothers. (46, p.79)

### First contact with HPs, assessment and provision of optimal care

Language difficulties (n=6) and a lack of shared decision-making (n=6) were barriers at these stages of the care pathway:

> When the midwife visits, I can only speak the sentences about requesting a translator … They said that this kind of service is limited … that is what is difficult being Chinese—language barrier. (47, p.6) [Language difficulties]

> … it would have been good I think to have been listened to about the side effects. I was on a very high dose of Olanzapine [sic] and it just knocks you out … (48, p.754) [Shared decision making]

### Organisational factors

Organisational-level factors were identified by 21 reviews.

### Screening/assessment

The most cited barriers to screening/assessment was the wording or contents of the tool (n=2), or if the tool was delivered in a tick-box way (n=6).

> There's so much more that you want to say, rather than just answering quite closed questions. (44, p.e695)

Some women found screening tools particularly problematic if the tool was not in her first language, indicating that cultural factors can overlap with organisational factors. For example, one review reported that certain questions may not elicit true feelings from Vietnamese women living in the UK because of the shame of admitting to these.[44] Further, question Q10 on the EPDS[45] ('the thought of harming myself has occurred to me') was seen as problematic to Arabic, Vietnamese and black Caribbean

mothers[44] living in the UK or USA, highlighting the need for culturally sensitive and relevant assessment tools.

### Access to care

Practical characteristics (n=5) of the organisation and services offered, such as a lack of childcare facilities, hard to reach locations and timing of appointments were a barrier to access:

> You have to have someone to look after your baby so who am I going to get to look after [my baby] (44, p.e695)

Other barriers at this stage of the care pathway included, a lack of services or overstretched services (n=7), a lack of collaboration across services (n=3) and lack of continuity of care (n=2):

> You shouldn't have to press that danger button of 'I'm gonna self-harm' or 'I'm gonna hurt my children' for someone to help you. (48, p.756) [Lack of services]

> My GP [general practitioner/family doctor] says go the HV [health visitor] and HV says go to GP. I don't know what to do, I need help, don't know where to go, or who to turn to (47, p.5) [Lack of collaboration across services]

> Every time I went to see the midwife, or…, I always had somebody different, and I don't want to tell 10 people my story. (48, p.752) [Lack of continuity of care]

Women reported wanting care that gave them an opportunity to talk to someone and discuss their emotional difficulties (n=8); some women wanted this opportunity within a peer support or group setting (n=12) and reported that an appropriate peer group could provide them with validation for their feelings (n=3). Care also needed to be individualised (n=10), and be culturally sensitive (n=8):

> In Pakistan we only saw lady professionals, but here you don't have a choice, you have to see the men as well otherwise you don't get to see a doctor… (51, p.10)

Women also appreciated care that provided them with information about PMH problems (n=5). Further, the location of the care needs to be easy to reach or carried out in women's homes (n=7), and women should not be discharged too early from these services (n=4).

### Political factors

Political factors were defined as factors that governmental agencies have influence over (eg, poverty, immigration, housing). Eight women identified these factors.

### Deciding to consult and access to care

Immigration status (n=4) and economic status (n=8) influenced women's decision to consult and access to care:

> Because when you're legal you can take the child to the day-care and look for a job… if you don't work, it's like you're dead, being alive… (52, p.13) [Immigration status]

> …if she has no money, how is she going to find help [with PPD]? (53, p.12) [Economic status]

This is due to the costs of healthcare and women's fear of being deported if they access help. Economic status was often exacerbated by immigration status with women reporting not being able to get health insurance due to their immigration status (n=4).

### Women's experience of care

Economic status also impacted women's experience of care in terms of women not being able to feel any sense of well-being when they were unable to fulfil basic needs such as '…*Not having enough money to make ends meet…*' (54, p.12) (n=4).

### Societal factors

Societal factors were identified in 24 reviews. The main societal factors that influenced a woman's journey along the care pathway were culture, societies' norms of what a 'good mum' should look like (maternal norms) and stigma. All these factors intertwine and influence one another. There was only one review that only included studies from LMICs,[46] therefore, these results mainly refer to western cultures.

### Deciding to consult and deciding to disclose

Stigma (n=21) and the maternal norm for women to show they are strong, that they can cope and be a good mother (n=18), prevented women from deciding to consult, and deciding to disclose:

> Mothers tend to think they should always be there. And mothers are supposed to be always rock solid, aren't they? Everyone assumes that. (56, p.568)

Adherence to cultural traditions (n=16) in women who had moved to Western countries, impacted their decision to consult and disclose. Two reviews reported that Hispanic women living in the USA felt they needed to remain strong (n=2), feeling they needed to show that they could cope, and that stigma prevented them from seeking help; they did not want to be seen as 'crazy' or 'loco' (57, p.97).

Four reviews found that South Asian women living in the UK did not consult or disclose for similar cultural reasons, for example, '*for fear of an inability to perform their role as a woman and a mother*' (58, p. 325), perceiving symptoms in religious terms '*All illness is coming from God*' (44, p.e649), and stigma:

> There is a huge stigma of being mentally ill in the public, but for us Asians there is a double disadvantage. I really fear that work will find out.' Pakistani woman living in the UK (47, p.5)

Black African and Caribbean women living in the UK or USA were also deterred from deciding to consult and disclose because of the expectation of women to be strong and be able to cope (n=4).

### Access to care, provision of optimal care and women's experience of care

Women's cultural backgrounds highlighted the need for culturally sensitive care. The lack of this care was as a barrier to access, provision of optimal care and women's experiences of care (n=8). Two reviews explained how Hispanic women living in the USA felt that language barriers, cultural insensitivity, and financial difficulties were a barrier to them accessing care.[47 48] Further, Jordanian women (living in Australia) spoke of being torn between their own cultural practices and Western health advice, having HPs putting pressure demands on them to change their beliefs and behaviours.[47] For women living in sub-Saharan Africa, the cultural tradition of confinement after birth meant women felt unable to leave their house for fear of being shamed. This was further exacerbated by the attribution of postnatal ill health to inadequate adherence to tradition.[46]

## DISCUSSION

This meta-review identified a wide range of barriers and facilitators to women accessing PMH care, that were influential at different levels (figure 2) and across different stages of the care pathway (figure 3).

Previous research has identified multiple factors that act as barriers to women seeking and accessing help for PMH problems. The factors include women not recognising the need to seek help,[49–53] the need for HPs to receive training on perinatal mental illness and cultural sensitivity,[44 49 54–60] continuity of care[49 51 56–59 61 62] and stigma.[47–49 53 55 56 60 61 63 64] Our findings are in line with these previous studies and adds to the body of evidence by identifying barriers and facilitators to PMH care, across the globe and presenting them on a multilevel model, and at different stages of the care pathway. This provides opportunities for HPs, service managers and policy-makers to identify barriers and facilitators that are most relevant to their context. The mapping of barriers and facilitators in this way, has also led to the development of evidence-based recommendations for design and delivery of PMH care.

### Recommendations for PMH care

The results from this meta-review can be used to inform healthcare providers and policy-makers on the optimal characteristics of PMH care and are summarised in table 1. This meta-review showed a complex interplay of multilevel factors that influence women's help-seeking and access to PMH care. Thus, recommendations for policy and practice also relate to both international-level guidelines, and guidelines for national and individual-level care. International-level guidelines should facilitate

more personalised care and should feed into national guidelines and be adopted where appropriate.

Societal factors such as stigma, maternal norms and culture play a large role in women accessing care. Research suggests that public mental health campaigns can increase knowledge about mental illness and improve attitudes about people with mental illness.[65] Therefore, increasing women's, families', those who have regular contact with women in the perinatal period, and the public's mental health literacy, could be carried out on an international level. This could be done through public health campaigns, and education within the community, such as antenatal education, and at healthcare appointments.

Maternal norms identified in this meta-review related to women believing that they needed to be strong and show they could cope. There may be some potential to change societal beliefs around maternal norms through increasing societal expectations about fathers' role in the family through more equal parental leave. For example, in countries where parental leave is more equal (eg, Finland), the uptake of paid paternity leave is higher.[66] Changing society's maternal norms could also be done by increasing women's equality. Research suggests that stereotypes of what a mother or a woman should look like are beginning to change in countries where women have gained more participation in the labour force[67] and have the right to access contraception and abortion.[68] However, research is needed to corroborate these findings.

At the political level, immigration and economic status, and healthcare costs were barriers to women accessing healthcare. The results from this meta-review show how race and gender interact to influence women's experiences of the healthcare system (intersectionality).[69] White women living within their country of birth who try to access PMH care are faced with barriers (eg, no childcare support), but women of colour, migrant women, or migrant women of colour are faced with additional barriers (eg, language barriers, structural/systematic discrimination). This finding is supported by research in general healthcare that has found ethnic minority and migrant women are disproportionately affected by existing barriers to accessing healthcare.[70] As found in this meta-review, these barriers include language and communication barriers, stigma, the cost of healthcare[71] and the inability to access culturally appropriate services.[72] This shows the need for equal rights to healthcare, regardless of immigration or economic status. Further, changes at the legislative level are needed to protect those who have migrated to a different country from being penalised for accessing healthcare.[71]

At the organisational level, this meta-review identified a range of factors that women viewed as ideal care. Women appreciated the opportunity to discuss screening results with HPs and for it not to be filled out as a 'tick-box' exercise.[73] In terms of treatment, women wanted the opportunity to talk to someone (a HP or a peer) about their difficulties.[53 56 57 69 72 74] They found peer support offered

them a sense of validation which they appreciated.[75] To overcome logistical barriers, the location of services should be easily accessible, or in women's homes.[44 48 56 61] Further, the length of treatment should be flexible and based on women's needs. Women did not want a 'one-size-fits-all' approach but wanted individualised treatment that was culturally appropriate.[47 49 54–57 59 75 76]

At the interpersonal and HP level, the characteristics of the HPs were important, as was their communication with women. Women reported that many HPs normalised their symptoms or were dismissive of their attempts to seek help. This may reflect the heavy workload experienced by many HP.[77–79] For example, research suggests that consultations where mental health problems are discussed take longer, and HPs often feel there is not enough time to address concerns fully.[79 80] This finding could also reflect inadequate training.[81] Within the UK, guidance states that all midwives and health visitors should receive training in order for them to identify, care for and refer women with PMH problems.[82] However, a synthesis of 30 studies found that midwives lack the confidence, knowledge and training to do this,[83] therefore, training around mental health is important. Another key training need is cultural sensitivity and cross-cultural understanding of PMH. Some systematic reviews in this meta-review identified that women were treated in a culturally insensitive way by HPs and that women of colour were less likely to be offered treatment or be asked about their mental health. It has been suggested that training given at medical and nursing school does not do enough to reduce unconscious biases against marginalised groups, which in turn influences treatment provided by healthcare providers.[84]

Improved interpretation services within PMH care may aid culturally sensitive care. Another potential way to improve culturally sensitive care is through the recruitment and retention of healthcare providers from diverse backgrounds.[85] This strategy has the potential to improve interpersonal relationships between HPs and patients,[86 87] which may therefore increase disclosure of PMH problems to HPs. In addition, research suggests increased representation of diverse populations in healthcare is associated with improved communication between health providers,[88 89] which therefore may reduce the risk of women falling through gaps in the care pathway.

Further, it has been argued that the way the western world views mental illness is very ethnocentric[90] and that culture and society influences what is viewed as a mental illness.[91] This may mean that some women's attempts to seek help are missed by HPs. It is, therefore, crucial that cultural sensitivity and cross-cultural mental health training is provided to HPs.

In terms of individual-level factors, many of these barriers can be improved through the recommendations suggested above. For example, improvement of knowledge around mental health is likely to reduce women's fear of judgement and self-stigma and increase her awareness of the symptoms she is experiencing, which may encourage help-seeking.[92] Redesign of care, such as providing easily accessible healthcare, may reduce the logistical barriers women experience.

## Strengths and limitations

The strength of this meta-review is the synthesis of a large amount of information from 32 systematic reviews from many different countries in order to identify barriers and facilitators to women deciding to seek help, access help and engage in PMH care. This information was then used to provide recommendations for the design and delivery of care. A limitation of the methodology is that only reviews published in academic journals and written in English language were included. Relevant reviews from health services, charities, third sector organisations, and other grey literature may have been missed. Another limitation is that only 10% of studies had duplicate data extraction. However, concordance was high, and it is therefore unlikely that any key themes were missed.

A limitation about the papers included in the meta-review was that most of them were rated as having low or critically low quality, meaning less confidence can be placed on their results. However, the qualitative sensitivity analysis found that most themes were supported in both the higher quality and lower quality reviews and including all reviews meant there was more focus on marginalised women, such as refugees, migrants and women living in sub-Saharan Africa. This shows that the results from this meta-review can be interpreted with reasonable confidence.

## Implications for future research

This review has revealed several limitations with the current evidence base on this topic. Very few systematic reviews (n=2) addressed the severity of illness, only one review looked at severe PMH problems[59] and most reviews (n=24) focused only on depression. There may be different barriers for other PMH problems therefore future research should focus on researching the barriers and facilitators to women with disorders other than depression. Another limitation with the identified reviews is that no reviews specified whether women had given birth to singletons only, or twins/higher-order multiples. This is important, as parents of twins or multiples report unique experiences in accessing PMH care.[93]

Furthermore, reviews only covered the inclusion of studies carried out in 25 countries, and only one review included studies that were only carried out in LMICs.[46] More research is needed in other countries to further aid our understanding of help-seeking and accessing care in women with PMH problems. In addition, none of the identified reviews included studies from diverse families, including same-sex couples and the transgender community. It is important that future research recruits more diverse populations to ensure all voices are heard.

Most reviews were rated as having low or critically low quality meaning less confidence can be placed on their results. However, the qualitative sensitivity analysis found that most themes were supported in both the higher

quality and lower quality reviews and including all reviews meant there was more focus on marginalised women, such as refugees, migrants and women living in sub-Saharan Africa. This shows that the results from this meta-review can be interpreted with reasonable confidence.

## Conclusion

The findings from this review point to a complex interplay of individual and system-level factors across different stages of the care pathway that can influence whether women seek help and access care for PMH problems. These factors should all be considered by policy-makers to improve the identification and treatment of PMH problems. Recommendations for the design and delivery of PMH care have been produced, building on the barriers and facilitators identified in this review. The identified barriers and facilitators point to the need for an international effort to reduce mental health stigma, and increase woman-centred, flexible care, delivered by well trained and culturally competent HPs.

**Author affiliations**
[1]Centre for Maternal and Child Health Research, City University, London, UK
[2]Department of Primary Care and Public Health, Brighton and Sussex Medical School, Brighton, UK
[3]Department of Women and Children's Health, School of Life Course Sciences, King's College London, London, UK
[4]Section of Women's Mental Health, Institute of Psychiatry, Psychology & Neuroscience, King's College London, London, UK
[5]Retired, Oxford, UK
[6]National Childbirth Trust, London, UK
[7]Bodleian Health Care Libraries, University of Oxford, Oxford, UK
[8]Nuffield Department of Population Health, National Perinatal Epidemiology Unit, Oxford, UK
[9]Nursing, Midwifery and Allied Health Professions Research Unit (NMAHP RU), University of Stirling, Stirling, UK
[10]Faculty of Education, University of Cambridge, Cambridge, UK

**Acknowledgements** We gratefully acknowledge the contribution of the MATRIx study team: Elaine Clark, Evelyn Frame, Simon Gilbody, Sarah McMullen, Camilla Rosan, Debra Salmon, Clare Thompson and Louise R Williams.

**Collaborators** The MATRIx Study Team: Elaine Clark, Evelyn Frame, Simon Gilbody, Sarah McMullen, Camilla Rosan, Debra Salmon, Clare Thompson, and Louise R Williams.

**Contributors** RW was involved in the design of the research and carried out screening, quality appraisal, analysis, write up of manuscript and editing of manuscript. NU contributed to screening and quality appraisal of papers and provided detailed feedback on the manuscript. GC contributed to screening and quality appraisal of papers provided detailed feedback on the manuscript. EF, AE and JS was involved in the conceptualisation of the project, the design of the research, analysis and provided detailed feedback on the manuscript. AH provided PPI input and detailed feedback on the manuscript. NR completed the literature searches. FA, AS, RC and SH contributed to the design of the research and provided detailed feedback on the manuscript. SA was the project manager, guarantor of the research, was involved in the conceptualisation of the project, the design of the research, analysis and provided detailed feedback on the manuscript. The MATRIx study team provided oversight to the running of the project, and feedback on the work produced.

**Funding** National Institute for Health Research and Health Services and Delivery Research Programme (NIHR128068).

**Competing interests** None declared.

**Patient and public involvement** Patients and/or the public were involved in the design, or conduct, or reporting, or dissemination plans of this research. Refer to the Methods section for further details.

**Patient consent for publication** Not applicable.

**Provenance and peer review** Not commissioned; externally peer reviewed.

**Data availability statement** Data are available on reasonable request. Data are available on reasonable request to the corresponding author.

**ORCID iDs**
Rebecca Webb http://orcid.org/0000-0002-8862-6491
Elizabeth Ford http://orcid.org/0000-0001-5613-8509
Abigail Easter http://orcid.org/0000-0002-4462-6537
Judy Shakespeare http://orcid.org/0000-0003-0770-8098

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
