## [Reviewer comments · BMJ Open]

ARTICLE DETAILS

TITLE (PROVISIONAL)	A meta-review of the barriers and facilitators to women accessing perinatal mental health care
AUTHORS	Webb, Rebecca; Uddin, Nazihah; Constantinou, Georgie; Ford, Elizabeth; Easter, Abigail; Shakespeare, Judy; Hann, Agnes; Roberts, Nia; Alderdice, Fiona; Sinesi, Andrea; Coates, Rose; Hogg, Sally; Ayers, Susan; Study Team, The MATRIx

VERSION 1 – REVIEW

REVIEWER	O'Mahony, Joyce Thompson Rivers University
REVIEW RETURNED	18-Sep-2022

GENERAL COMMENTS	I enjoyed reading the manuscript as the topic and findings are very relevant to the scope of the BMJ Open. It is a topic that needs more awareness and attention to advance knowledge concerning perinatal mental health support and services for women. The following recommendations are offered to strengthen your paper as a scholarly contribution: Overall, I thought the work to produce this review article was commendable and is very much needed as we grapple globally with this underscored area in women's perinatal health. Organization Firstly, the authors have clearly presented the steps of the systematic review process with the outcome of 32 articles. This is evident throughout the review. Pg 6 line 8-9 I understand what you are stating but could this point be introduced later. I believe there are more pressing reasons why Perinatal MH needs attention. Pg 9 line 22 You briefly introduce "Each level of the multi-level framework Figure 2" In my view, this framework strongly resembles the socioecological model/framework. Pg 37 mapped across a multi-level model of influential factors (individual, healthcare professional, interpersonal, organizational, political, and societal)? This needs to be acknowledged/cited. Can you provide more description about this framework-connection and intersectional elements. For example, 'culture' can be threaded and identified throughout many levels in your framework. I was also surprised that 'gender/role of' was only briefly touched on in your writing. This is, I believe is an oversight of your work. You mention in your recommendation pg 12: "The continuation of
--

	international policies to promote gender equality” This is vague/superficial- there is so much more to add here. Then pg 17 line 33 you mention “e.g. the importance of fulfilling traditional gender and maternal roles, perceiving symptoms in religious...” This is heightened for immigrant/refugee populations. You may want to recheck your studies. P 20 line 40 “Another key training need is cultural sensitivity and cross-cultural of PMH” This recommendation is not new and now has been reported for numerous years. Outlining what we already know about PMH (and recommendations) and what your paper adds is an excellent way to highlight the findings of your review. In its present form however, there is more analysis and work to be conducted from your review of studies. At minimum reread your findings and discussion to determine what this meta-review review adds. I hope the authors choose to develop this paper further, as it has potential to make an important contribution with improvements.
--	---

REVIEWER	Wenze, Susan Lafayette College, Psychology
REVIEW RETURNED	19-Jan-2023

GENERAL COMMENTS	Thank you for the opportunity to review this paper. This manuscript is a meta-review of 32 review articles examining barriers & facilitators to women accessing perinatal mental health care, spanning diverse geographic areas & cultural backgrounds. The authors argue that each included systematic review is different in terms of aims & methods, so a meta-review is useful to draw overall conclusions & summarize this literature. While understanding why women do or do not access perinatal mental health care is critical, combining information from diverse & variable studies to yield meaningful conclusions is challenging, particularly when the quality is low for most of them (as is the case in this paper). Nevertheless, this paper aims to do so. Overall, it is well-written & clear &, as noted, it tackles an important topic. It also includes data from woefully under-studied areas (e.g., LMICs). All that said, I have some suggestions that I believe would strengthen the claim that the manuscript makes a necessary & novel contribution to the literature on perinatal mental health care. 1. My primary question/concern has to do with the somewhat conflicting arguments that the authors make throughout the paper about the need for tailored, culturally-specific care vs. universal, international guidelines. On the one hand, existing reviews of PMH services, barriers, & facilitators vary in their methodology, included populations, etc. And one of the main conclusions of this paper is that tailored, culturally-appropriate interventions are needed to circumvent different populations’ unique barriers & facilitators to care (women don’t want “one size fits all” approaches). At one point, the authors even acknowledge that different kinds of care might or might not be available or feasible in one geographic area vs. another. But on the other hand, they also argue that an overarching summary that is internationally applicable is necessary & this is the key aim of this paper. There is probably a
---

	way to reconcile these conflicting claims but I don't think the paper adequately does so in its present form. 2. On a (semi-)related note, I'm not sure that I follow the argument that, "As most PMH services are designed for all women within a population, regardless of their background, a summary of all the literature is needed." How does the need for a summary (which arguably will result in overarching conclusions, rather than a fine-grained analysis of what kinds of services will work best for specific subpopulations) follow from the assertion in the first part of this sentence? 3. As a reader, I wanted more details & context in the Introduction. For example, the link between PMH problems & adverse outcomes is mentioned, but I wanted some examples. What kinds of adverse outcomes? What kinds of help-seeking barriers? By "PMH," do the authors mean threshold-level diagnoses? Sub-threshold symptoms? Stress? In the PROSPERO Register in the Appendix, the authors say PMH encompasses anxiety, depression, stress, & adjustment disorders. Later, they also mention PTSD. And some of the included studies also looked at things like bipolar disorder & panic disorder. 4. How were differences of opinion resolved (Synthesis of Results)? 5. The fact that "Most reviews were evaluated as having low (n = 14) or critically low (n = 5) confidence with their results" should be discussed in the Limitations section. 6. Were the included studies focused on women with singletons? Women with twins or higher-order multiples often report unique postpartum experiences, including barriers & facilitators to PMH care. 7. Table 1 could be better organized. Barriers & facilitators are mixed together in no logical order. 8. In Table 1, is "Ideal Care" meant to be a separate subheading? This is not a level of care in the same way that the other levels (interpersonal, organizational, societal, etc.) are. 9. It seems that Table 2 should be organized in the same order (individual to societal levels) as Table 1. Currently, these two tables present the levels in opposite orders. 10. I find myself struggling with the organization of Table 2. For example, how are "interpersonal" & "healthcare professional" different? Wouldn't the themes that load on each overlap or even be identical? How is "continuity of carer" an interpersonal theme? Isn't that something that the organization (not the individual provider) has control over? And if it IS a theme at the interpersonal level, shouldn't there be a corresponding individual-level recommendation to promote continuity of care? Shouldn't "free access to healthcare" & "woman-centered care" be political or organizational (not individual) recommendations? 11. Under Results, Organizational Factors, the examples did not seem to follow from the assertion that "Some women found screening tools particularly problematic if the tool was not in her first language." The examples (Vietnamese women living in the UK; Arabic, Vietnamese, Black Caribbean mothers living in the UK or USA) seem to capture stigma & cultural sensitivity/relevance, not a language barrier. 12. The end of the first full Discussion paragraph proposes actions that "should be carried out on an international level," but I think this would look quite different in different locations, no? Indeed, shortly thereafter (page 20), the authors report that "Women did not want a 'one size fits all' approach but wanted individualized treatment that was culturally appropriate." This gets back to my first
--	--

	comment, above: How to balance international guidelines & recommendations w/ the clear need for individualized, tailored, culturally- & locally-appropriate care. 13. In the Discussion section, the authors rightly report that cultural sensitivity & cross-cultural mental health training for HPs might help address some of the concerns raised. An additional route might be to increase the pipeline & improve retention of diverse providers. A sentence or two on the importance of this (difficult but critical) strategy would be useful. 14. Can the authors suggest ways to harness other facilitators to accessing care? I'm thinking in particular about ways to educate friends, family, or even key community members (religious leaders, teachers, etc.) about ways to improve rates of women accessing care, since "supportive family & friends" was listed earlier as a facilitating factor. 15. In two places, the numbers in the CONSORT flow chart do not add up. (1) EndNote automatic removal of irrelevant studies (n = 66) is comprised of fetal distress (n = 46), oxidative distress (n = 9), & non-English (n = 66). (2) Full text articles assessed for eligibility (n = 66) minus Full-text articles excluded (n = 34) should equal 32, but the next box states that 34 studies were included. The abstract & elsewhere in the paper notes that 32 studies were included. 16. I am not suggesting that they incorporate such papers, but did the authors do anything to at least try to determine whether - & how many - reviews published in languages other than English exist? Minor comments:  1. In the Abstract, I believe the sentence should read, "...highlighting the need for an international effort to increase awareness of PMH *problems*, reduce mental health stigma..." 2. The first sentence of the Introduction is missing the word "with" (i.e., "Perinatal mental health (PMH) problems are associated *with* adverse outcomes for women..."). 3. Who (which author[s]) conducted searches (Method)? This information is listed for the other tasks. 4. On page 20, the authors claim that HPs normalizing or dismissing symptoms or attempts to seek care could be a reflection of inadequate training. Lack of resources (including time!) should probably be mentioned here, too. Unfortunately, a busy provider who is working a 10 to 12-hour day without breaks might be dismissive because they don't have the bandwidth to deal with extra tasks during their work. 5. The last sentence of the conclusion would be clearer with a slight word adjustment: "The identified barriers & facilitators point to the need for an international effort to reduce mental health stigma and *INCREASE* woman-centered, flexible care...." 6. This is not necessarily something to change, but I was surprised that lack of parental leave didn't come up as a barrier to accessing care. At least in the United States, this is often a big barrier to accessing care among new parents. But I suppose the fact that it wasn't a common theme at any level is a good thing!
--	---

VERSION 1 – AUTHOR RESPONSE

Reviewer: 1 Dr. Joyce O'Mahony, Thompson Rivers University Comments to the Author:

I enjoyed reading the manuscript as the topic and findings are very relevant to the scope of the BMJ Open. It is a topic that needs more awareness and attention to advance knowledge concerning perinatal mental health support and services for women. The following recommendations are offered to strengthen your paper as a scholarly contribution. Overall, I thought the work to produce this review article was commendable and is very much needed as we grapple globally with this underscored area in women's perinatal health.

Thank you for the positive feedback on our manuscript. We too agree this is a topic that deserves more attention.

Organization

Firstly, the authors have clearly presented the steps of the systematic review process with the outcome of 32 articles. This is evident throughout the review.

Thank you for your positive feedback.

Pg 6 line 8-9 I understand what you are stating but could this point be introduced later. I believe there are more pressing reasons why Perinatal MH needs attention.

Thank you for pointing this out. We have extended the first paragraph about the impacts of PMH on women and their families in the introduction (P 6-7)

Pg 9 line 22 You briefly introduce "Each level of the multi-level framework Figure 2" In my view, this framework strongly resembles the socioecological model/framework. Pg 37 mapped across a multi-level model of influential factors (individual, healthcare professional, interpersonal, organizational, political, and societal)? This needs to be acknowledged/cited.

Can you provide more description about this framework- connection and intersectional elements. For example, 'culture' can be threaded and identified throughout many levels in your framework.

This framework is first introduced in the synthesis of results section, P9 "Themes were mapped...".

We have now added more detail to this section, and more details to the section "determining the barriers and facilitators to women..." on p10.

I was also surprised that 'gender/role of' was only briefly touched on in your writing. This is, I believe is an oversight of your work. You mention in your recommendation pg 12: "The continuation of international policies to promote gender equality" This is vague/superficial- there is so much more to add here.

Thank you for pointing this out. There is more detail about this recommendation in the discussion on P22, but we have now added it to Table 2 to make this clearer.

Then pg 17 line 33 you mention "e.g. the importance of fulfilling traditional gender and maternal roles, perceiving symptoms in religious..." This is heightened for immigrant/refugee populations. You may want to recheck your studies.

We have provided more context to this statement (P19) with quotes from relevant papers, and a reference to Quote 21 which is in Table 1. Furthermore, there is a discussion of these factors in the discussion on p22 where we discuss maternal norms, laws that have been associated with more equality, and intersectionality.

P 20 line 40 "Another key training need is cultural sensitivity and cross-cultural of PMH"

This recommendation is not new and now has been reported for numerous years.

Outlining what we already know about PMH (and recommendations) and what your paper adds is an excellent way to highlight the findings of your review.

In its present form however, there is more analysis and work to be conducted from your review of studies. At minimum reread your findings and discussion to determine what this meta-review review adds.

Thank you for your feedback. We have added a paragraph at the beginning of the discussion P20-21 to explain what we already knew about perinatal mental health (including the need for training), and what this paper adds.

Reviewer: 2 Dr. Susan Wenze, Lafayette College

Thank you for the opportunity to review this paper. This manuscript is a meta-review of 32 review articles examining barriers & facilitators to women accessing perinatal mental health care, spanning diverse geographic areas & cultural backgrounds. The authors argue that each included systematic review is different in terms of aims & methods, so a meta-review is useful to draw overall conclusions & summarize this literature. While understanding why women do or do not access perinatal mental health care is critical, combining information from diverse & variable studies to yield meaningful conclusions is challenging, particularly when the quality is low for most of them (as is the case in this paper). Nevertheless, this paper aims to do so. Overall, it is well-written & clear &, as noted, it tackles an important topic. It also includes data from woefully under-studied areas (e.g., LMICs). All that said, I have some suggestions that I believe would strengthen the claim that the manuscript makes a necessary & novel contribution to the literature on perinatal mental health care.

Thank you for your positive comments and your detailed feedback below. We have addressed this and believe it makes the manuscript stronger.

1. My primary question/concern has to do with the somewhat conflicting arguments that the authors make throughout the paper about the need for tailored, culturally-specific care vs. universal, international guidelines. On the one hand, existing reviews of PMH services, barriers, & facilitators vary in their methodology, included populations, etc. And one of the main conclusions of this paper is that tailored, culturally-appropriate interventions are needed to circumvent different populations' unique barriers & facilitators to care (women don't want "one size fits all" approaches). At one point, the authors even acknowledge that different kinds of care might or might not be available or feasible in one geographic area vs. another. But on the other hand, they also argue that an overarching summary that is internationally applicable is necessary & this is the key aim of this paper. There is probably a way to reconcile these conflicting claims but I don't think the paper adequately does so in its present form.

We have provided more context about what we mean by international guidelines in the introduction (P6-7). We understand and recognise this conflict, and have provided more information in the discussion (P21) and Table 2 which states that international policy needs to support the implementation of more personalised care.

2. On a (semi-)related note, I'm not sure that I follow the argument that, "As most PMH services are designed for all women within a population, regardless of their background, a summary of all the literature is needed." How does the need for a summary (which arguably will result in overarching conclusions, rather than a fine-grained analysis of what kinds of services will work best for specific subpopulations) follow from the assertion in the first part of this sentence?

Thank you for pointing this out. This sentence was not clear, and we have now removed it from the introduction. We have replaced it with a sentence regarding the benefits of conducting meta-reviews as stated by Grant and Booth (2009) and Samnani et al. (2017).

3. As a reader, I wanted more details & context in the Introduction. For example, the link between PMH problems & adverse outcomes is mentioned, but I wanted some examples. What kinds of adverse outcomes? What kinds of help-seeking barriers? By "PMH," do the authors mean threshold-level diagnoses? Sub-threshold symptoms? Stress? In the PROSPERO Register in the Appendix, the authors say PMH encompasses anxiety, depression, stress, & adjustment disorders. Later, they also mention PTSD. And some of the included studies also looked at things like bipolar disorder & panic disorder.

Thank you for this point. We have added more context in the introduction about PMH difficulties, such as how they can refer to adjustment disorders (e.g., sub-threshold symptoms) as well as more severe difficulties such as postpartum psychosis (P6). We have also provided more information about the adverse impacts of PNMH difficulties (P-7)

4. How were differences of opinion resolved (Synthesis of Results)?

Thank you for pointing out the omission of this. We have now added this to P9.

5. The fact that “Most reviews were evaluated as having low (n = 14) or critically low (n = 5) confidence with their results” should be discussed in the Limitations section.

This is a very good point, we have added this into the limitations section (P25).

6. Were the included studies focused on women with singletons? Women with twins or higher-order multiples often report unique postpartum experiences, including barriers & facilitators to PMH care.

None of the reviews provided this detailed level of information about the included studies. We have added this to the discussion as a limitation (P25)

7. Table 1 could be better organized. Barriers & facilitators are mixed together in no logical order. Thank you for your point. The order they are presented in relate to the order they relate to in the text. They have been presented in a table, rather than in text due to word limit. We have now added quote numbers in the text (p16-20) and on Table 1 so these can be cross referenced.

8. In Table 1, is “Ideal Care” meant to be a separate subheading? This is not a level of care in the same way that the other levels (interpersonal, organizational, societal, etc.) are.

We have now added an explanation for this on P10.

9. It seems that Table 2 should be organized in the same order (individual to societal levels) as Table 1. Currently, these two tables present the levels in opposite orders.

Thank you for pointing this out. We have re-organised this table.

10. I find myself struggling with the organization of Table 2.

We have broken this point down so we can answer your individual questions

For example, how are “interpersonal” & “healthcare professional” different? Wouldn't the themes that load on each overlap or even be identical?

We have now provided a more detailed explanation of the model on P10 which explain the differences and how they overlap onto one another.

How is “continuity of carer” an interpersonal theme? Isn't that something that the organization (not the individual provider) has control over? And if it IS a theme at the interpersonal level, shouldn't there be a corresponding individual-level recommendation to promote continuity of care?

Since the submission of this manuscript in July 2022, we have had meetings with stakeholders (i.e. health professionals, service managers, women and families). These discussions have meant that continuity of carer has indeed been moved to organisational. It was originally there because continuity of carer helps to promote a good relationship and rapport between women and families. We have also explained on Table 2 that the recommendation isn't necessarily for that group to resolve, but for another group (see footnotes).

Shouldn't “free access to healthcare” & “woman-centered care” be political or organizational (not individual) recommendations?

We have also explained on Table 2 that the recommendation isn't necessarily for that group to resolve, but for another group (see footnotes).

11. Under Results, Organizational Factors, the examples did not seem to follow from the assertion that "Some women found screening tools particularly problematic if the tool was not in her first language." The examples (Vietnamese women living in the UK; Arabic, Vietnamese, Black Caribbean mothers living in the UK or USA) seem to capture stigma & cultural sensitivity/relevance, not a language barrier.

We hope that this has been resolved by the definition of the model we have provided in the section "determining the barriers and facilitators..." on P10, this states that anything related to the choices/delivery of the care that is provided is an organisational level factor. We have also clarified that this organisational level barrier overlaps with cultural factors.

12. The end of the first full Discussion paragraph proposes actions that "should be carried out on an international level," but I think this would look quite different in different locations, no? Indeed, shortly thereafter (page 20), the authors report that "Women did not want a 'one size fits all' approach but wanted individualized treatment that was culturally appropriate." This gets back to my first comment, above: How to balance international guidelines & recommendations w/ the clear need for individualized, tailored, culturally- & locally-appropriate care.

We have provided an explanation that international guidelines should facilitate personalised care in the discussion (P21), and in Table 2.

13. In the Discussion section, the authors rightly report that cultural sensitivity & cross-cultural mental health training for HPs might help address some of the concerns raised. An additional route might be to increase the pipeline & improve retention of diverse providers. A sentence or two on the importance of this (difficult but critical) strategy would be useful.

Thank you for this suggestion. We have added it in to the discussion to p.24.

14. Can the authors suggest ways to harness other facilitators to accessing care? I'm thinking in particular about ways to educate friends, family, or even key community members (religious leaders, teachers, etc.) about ways to improve rates of women accessing care, since "supportive family & friends" was listed earlier as a facilitating factor.

We have listed the improvement of public mental health literacy as a recommendation in the discussion (P20-21), in order to educate friends and family about perinatal mental health. We have clarified in Table 2 and the discussion that this should not just be targeted at perinatal women.

15. In two places, the numbers in the CONSORT flow chart do not add up. (1) EndNote automatic removal of irrelevant studies (n = 66) is comprised of fetal distress (n = 46), oxidative distress (n = 9), & non-English (n = 66). (2) Full text articles assessed for eligibility (n = 66) minus Full-text articles excluded (n = 34) should equal 32, but the next box states that 34 studies were included. The abstract & elsewhere in the paper notes that 32 studies were included.

Thank you for pointing this out. We can confirm that only 32 studies were included, not 34. We have updated the PRISMA flow diagram to reflect this.

16. I am not suggesting that they incorporate such papers, but did the authors do anything to at least try to determine whether - & how many - reviews published in languages other than English exist?

Unfortunately not, we made the decision early to exclude non-English papers due to the costs of translation. We have explained this on P8 and it is noted as a limitation in the discussion (p25).

Minor comments:

1. In the Abstract, I believe the sentence should read, "...highlighting the need for an international effort to increase awareness of PMH *problems*, reduce mental health stigma..."

Thank you for pointing this out. We have added the word difficulties here (p.3)

2. The first sentence of the Introduction is missing the word "with" (i.e., "Perinatal mental health (PMH) problems are associated *with* adverse outcomes for women...").

We have now added more information to the introduction meaning this sentence has been removed.

3. Who (which author[s]) conducted searches (Method)? This information is listed for the other tasks.

Thank you for pointing this out. We have now added this (P8).

4. On page 20, the authors claim that HPs normalizing or dismissing symptoms or attempts to seek care could be a reflection of inadequate training. Lack of resources (including time!) should probably be mentioned here, too. Unfortunately, a busy provider who is working a 10 to 12-hour day without breaks might be dismissive because they don't have the bandwidth to deal with extra tasks during their work.

Thank you for pointing this out. We have now added this to P23.

5. The last sentence of the conclusion would be clearer with a slight word adjustment: "The identified barriers & facilitators point to the need for an international effort to reduce mental health stigma and *INCREASE* woman-centered, flexible care..."

Thank you for pointing this out. We have now added this (P26).

6. This is not necessarily something to change, but I was surprised that lack of parental leave didn't come up as a barrier to accessing care. At least in the United States, this is often a big barrier to accessing care among new parents. But I suppose the fact that it wasn't a common theme at any level is a good thing!

The closest theme to this is practical barriers to attending care, which includes not having anyone to look after the baby. Only three reviews specifically mentioned parental leave, and two of the reviews cite the same study by Morrow et al. (2008)

1. "We have shared [household tasks] even if I have done most of them. . . then I think. . . that I also feel that it is still the woman who is the head parent somehow. . . Fathers should not whine, nor be in the center. . . they should be happy because they had become fathers and can take parental leave."39(p.434) Holopainen & Hakulinen

2. Morrow et al. (2008) also found poverty resulting specifically from insecure employment and immigration status influenced experiences of social support among immigrant and refugee women with PPD. These specific factors contributed to increased fear of partners losing income if they requested parental leave. This fear left women with unmet emotional needs and limited choices in accessing social support (Callister, Beckstrand & Corbett, 2011; Morrow et al., 2008; O'Mahony, Donnelly, Este, et al., 2012). – Kassam (2019)

3. In this situation, women felt they could not ask their partners to take any time off work to be with them and to help them care for their baby. In the study by Morrow [32], women spoke about their husbands' long working hours and that they were often alone following birth because their husband had to be at work, I told him he could request parental leave, but he was worried that the company would fire him, so he didn't ask for parental leave. So nobody took care of me.([32], p.604) – Schmieid et al. (2017)

REVIEWER	Wenze, Susan Lafayette College, Psychology
REVIEW RETURNED	31-Mar-2023

GENERAL COMMENTS	Thank you for the opportunity to review a revised version of this manuscript. The authors have fully addressed all of my original suggestions, comments, and concerns, and I believe the article is ready for publication. I look forward to seeing this important work in print!
---

VERSION 2 – AUTHOR RESPONSE

Reviewer: 2 Dr. Susan Wenze, Lafayette College Comments to the Author:

Thank you for the opportunity to review a revised version of this manuscript. The authors have fully addressed all of my original suggestions, comments, and concerns, and I believe the article is ready for publication. I look forward to seeing this important work in print!

Thank you very much for your positive feedback on our manuscript. We want to thank you for your time in reviewing it again.